# Non-Classical Model of Dynamic Behavior of Concrete

**Adam Stolarski \*** **, Waldemar Cichorski \*** **and Anna Szcześniak**

Faculty of Civil Engineering and Geodesy, Military University of Technology, 2 gen. Sylwestra Kaliskiego Street, 00-908 Warsaw, Poland

\* Correspondence: adam.stolarski@wat.edu.pl (A.S.); waldemar.cichorski@wat.edu.pl (W.C.); Tel.: +48-261-839-587 (A.S.); +48-261-837-784 (W.C.)

**Featured Application: The dynamic analysis of reinforced concrete structures, and the prognosis of the damage development of reinforced concrete structural elements.**

**Abstract:** Modeling of dynamic properties of concrete is presented in the paper. The non-classical model of dynamic deformation was proposed. The essence of this model is the method of determination of the initial dynamic yield surface. For this purpose, the dynamic strength criterion was used. The model describes the elastic properties until attaining the dynamic strength of concrete, perfectly plastic properties in the limited range of deformation, material softening, material dilatation, and cracking or crushing of material as the residual stress processes during tension or compression. Degradation of elastic material constants was taken into consideration. Comparative analysis with previously published experimental results and theoretical models demonstrated that the proposed model is well approximates the basic dynamic properties of concrete and can be used in numerical analysis to evaluate the dynamic load capacity of reinforced concrete structures.

**Keywords:** dynamic models; concrete; constitutive relations; cracking; crushing; model verification

## 1. Introduction

Modeling of properties of concrete for the purpose of stress analysis of engineering structures can be related to the so-called macroscopic level in which concrete can be treated—at least in the initial phase of deformation—as isotropic and homogeneous material. Taking into consideration such an assumption enables the use of the phenomenological description of concrete behavior as a solid. Nevertheless, the constitutive model should describe the basic properties of the material. Detailed information in the range of physical properties of concrete observed in the static experiments are given among others in the papers of Kupfer et al. [1], Mills and Zimmerman [2], Luanay and Gachon [3], Schickert and Winkler [4], and Tasuji et al. [5]. The results of these and other papers are cited and commented also in the studies of Nilsson [6], Klisiński [7], Podgórski [8], and Hofstetter and Mang [9].

Experimental results concerning the dynamic properties of concrete are concentrated mainly on the investigation of the relationship between dynamic and static strength under different types of loads, (Watstein [10], Hansen et al. [11], Bazenov [12], Zieliński [13]).

Analytical relationships of dynamic strength hardening coefficients on strain rates, and the approximating results of experimental investigations are also included in the papers of Nilsson [6], Bazenov [12], and Zieliński [13] as well as in the papers of Rostasy and Hartwich [14], Soroushian et al. [15], and Popov [16].

Apart from dynamic strength hardening, the deformation response of concrete is characterized by the modulus of deformation. In the papers of Watstein [10], Bazenov [12], and Soroushian et al. [15], increasing the dynamic modulus of deformation in proportion to a static one was shown on the base

of experiments. However, coefficients increasing the dynamic modulus of deformation are less than coefficients increasing the dynamic strength hardening of concrete.

Full, dynamic stress–strain relationship, especially concerning post critical material softening is the subject of a few papers. In a paper by Watstein [10], the dynamic stress–strain curves are described illustrating the behavior of concrete up to attaining dynamic strength. Typical, normalized dynamic stress–strain curves describing material softening paths of deformation are included in the papers of Rostasy and Hartwich [14], Dilger et al. [17], and Kowalczyk and Dilger [18]. From application in the dynamics point of view, description of the behavior of concrete in the unloading and reloading processes has a very important meaning. However, experimental results for concrete under cyclic loading only are attainable in the literature (e.g., Karsan and Jirsa [19], Sinha et al. [20]). Despite that, these investigations do not include real dynamic loads, and quality information from these experiments can be applied in simplified modeling of dynamic unloading and reloading processes.

Well-known constitutive models, in which different ranges of the specific concrete features are taken into consideration, are often adapted in the description of the behavior of concrete. In relation to the static behavior of concrete, it is possible to indicate the exemplified application of the theory of hypoelasticity (e.g., Nilson [21]), theory of infinitesimal elastic plastic deformation (e.g., Bažant and Tsubaki [22]), theory of hyperelasticity (e.g., Kupfer and Gerstle [23]), theory of incremental elastic–plastic yielding (e.g., Willam and Warnke [24]), theory of degradation (e.g., Bažant and Kim [25], Dragon and Mróz [26], Klisiński [7]), and endochronic theory of viscoplasticity (e.g., Bažant and Bhat [27]).

In the literature, a few papers are known from the range of modeling the dynamic behavior of concrete. Theoretical investigations are presented in a paper by Kowalczyk and Sawczuk [28], in which the general structure of constitutive relations and failure criterion for concrete are described in the form of representation of tensor functions.

The most complete and useful for practical application is the model of dynamic deformation of concrete presented by Nilsson [6]. This model describes the elastic-viscoplastic-plastic-brittle-material softening features of concrete and it constitutes the specific application of Perzyna's elastic-viscoplasticity theory (Perzyna [29]).

Investigations in the field of dynamic modeling of concrete behavior are the subject of many scientific considerations.

Computational analysis of the influence of boundary conditions on the behavior of concrete under dynamic load, described using well-known constitutive models of material, was presented by Marzec and Tajchman [30]. The 3D damage model for concrete under dynamic load that includes strain rate effects and confinement effects was presented in Mazars et al. [31].

The model of concrete describing the dynamic failure of material under multiaxial loading was presented in Grassl et al. [32]. The nonlinear dynamic multiaxial strength criterion for concrete was presented in Wang et al. [33]. The criterion was obtained by transforming the static multiaxial strength criterion to a dynamic criterion by building relationships between the material parameters of the static criterion and strain rate. A new material model for concrete under intense dynamic loading was presented by Kong et al. [34]. This model is implemented into the commercial finite element code.

Modeling of specific properties of concrete under the dynamic loads such as fragmentation of material was presented in Forquin and Erzar [35] or the fracture of material which was presented by Ožbolt et al. [36] and Snozzi et al. [37].

The papers concerning the modeling of dynamic behavior of concrete under uniaxial compression form a separate group of papers. A paper by Soroushian et al. [15] presented a model which described the nonlinear deformation up to attaining the dynamic strength and linear material softening. In turn, Popov [16] presented the approximation of the increasing path of a dynamic stress–strain diagram in the bisectional form consisting of linear and power curves. On the other hand, in papers by Bąk and Stolarski [38,39], the nonstandard three-sectional model of dynamic deformation of concrete under uniaxial compression was presented.

Applying the conclusions from static multiaxial tests results and dynamic tests results for uniaxial compression, the dynamic model for concrete in case of complex stress states is proposed in the paper. The model can be characterized as the elastic–plastic modified in the non-classical way by introducing the dynamic strength criterion by Stolarski [40], for the purpose of the initial dynamic yield surface determination.

The novelty of the proposed model is the connection of two different stages of the deformation process. Namely, process of achieving the dynamic strength of concrete, and quasi – static incremental elastic – plastic process with the initial condition at the level of determined dynamic strength.

In the description of the elastic – plastic process, the model shows some similarity to the concrete plastic damage model (see: Lubliner et al. [41], Lee and Fenves [42]). In the proposed model, the adopted evolution law for the yield surface in the range of material softening can be used to determine the degradation / softening parameter similar to the scalar damage parameter. This fact was used in the description of the degradation of the concrete deformation modulus.

Because the comparative analysis with previously published experimental results and theoretical models shoved that the proposed model well approximates the basic dynamic properties of concrete, so it can be useful in numerical analysis to evaluate the dynamic load capacity of reinforced concrete structures. Moreover, the proposed model can be suitable for the post– critical analysis of the dynamically loaded reinforced concrete elements as well as for predicting the development of structural elements failure and finally for allowing a better analysis of the construction systems' safety.

## 2. Model of Concrete

### 2.1. General Concept of the Model

The idea of a non-classical model of dynamic behavior of concrete consists in combining two stages of the deformation process. Stage 1 is the process of elastic dynamic deformation, in which the entire process of reaching dynamic strength of concrete is cumulated. Stage 2 is the process of elastic–plastic deformation of the material, which is described by the dynamic strength determined in stage 1.

The proposed model is a four-phase approximation of the nonlinear behavior of concrete. Particular phases describe the elastic properties of concrete, its limited capabilities for deformation in the range of plastic flow on the initial, dynamic yield surface, material softening, and the residual (stress-free) state. The basic element of the model is the method of determination of the initial dynamic yield surface. For this purpose, the dynamic strength criterion was used (Stolarski [40]). The criterion includes the sensitivity of concrete to the stress history in the initial range of deformation idealized as an elastic process. The calculated dynamic strength determines by material constants the initial, dynamic yield surface of concrete as an elastic–plastic material with further stressing processes running independent of the strain rate. For the purpose of the plastic part of strain-rate determination, the non-associated flow rule was postulated. The plastic potential function was introduced as the modified yield surface function including an additional material constant, which makes possible the control of material volume change during the plastic deformation process. The material softening phase is modeled as the plastic flow on the transient yield surface. The softening parameter is dependent on the effective plastic strains and the softening modulus controls the process of isotropic shrinkage of the yield surface. Degradation of the elastic material constants was included, but the unloading/reloading processes are assumed as linear and elastic. The description of the plastic flow process in the material softening phase requires the application the less rigorous Il'Yushin postulate of positive plastic power in place of the Drucker postulate of material stability [43]. The model enables the simplified description of the smeared cracking or crushing processes which are concentrated in the regions of the tensile or compressive residual stress states.

## 2.2. Yield Surface

Properties of limit surface, especially its good agreement with experimental results for concrete in complex stress states, are proposed by Stolarski [40], and give reason for applying the following equation as a yield surface for the elastic–plastic material:

$$F\left(\sigma_{ij}, K\right) = \left[\frac{\tau_0}{\rho(\varphi)} + aK\right]^2 - bK\sigma_0 - cK^2 = 0 \tag{1}$$

where $\tau_0 = \sqrt{2J_2/3}$ is the tangent octahedral stress; $\sigma_0 = I_1/3$ is the mean normal stress; $a$, $b$, $c$ are the material constants, dependent on the basic strength of concrete in uniaxial and biaxial compression and uniaxial tension, and $K$ is the evolution parameter.

The function $\rho(\varphi)$, determining the shape of the yield surface cross-section by the octahedral plane $\sigma_0 = const$, is assumed in the form proposed by Willam and Warnke [24]:

$$\rho(\varphi) = \frac{2\left(1 - \lambda^2\right)\cos\left(\frac{\pi}{3} - \varphi\right) + (2\lambda - 1)\sqrt{4(1 - \lambda^2)\cos^2\left(\frac{\pi}{3}\right) + 5\lambda^2 - 4\lambda}}{4(1 - \lambda^2)\cos^2\left(\frac{\pi}{3} - \varphi\right) + (2\lambda - 1)^2} \tag{2}$$

where $\varphi$ is the angle dependent on the stress state:

$$\cos 3\varphi = \sqrt{2}\frac{J_3}{\tau_0^3} \tag{3}$$

In Equations (1—3), the following denotations are introduced: $I_1 = \sigma_{ii}$ is the first invariant of stress tensor $\sigma_{ij}$; $J_2 = (1/2)s_{ij}s_{ij}$ and $J_3 = (1/3)s_{ij}s_{jk}s_{ki}$ are the second and the third invariants of stress deviator $s_{ij} = \sigma_{ij} - (1/3)\sigma_{kk}\delta_{ij}$; $i$, $j$, $k$ = 1, 2, 3.

Parameter $\lambda$ determines the relation between characteristic values of the radius $\rho(\varphi)$ for uniaxial tension or biaxial compression $\rho_t$ and for uniaxial compression $\rho_c$:

$$\lambda = \frac{\rho_t}{\rho_c} \tag{4}$$

and can be recognized as the material feature.

The constant value of parameter $\lambda$, determining the so-called tension curve in the equation of yield surface was assumed on the basis of Reimann's definition, which agrees with the experimental results of Reimann [44]:

$$\lambda = \frac{3\varphi_{cc}\varphi_t + \varphi_{cc} - \varphi_t}{2\varphi_{cc} - \varphi_t} \tag{5}$$

where non-dimensional static strengths of concrete

$$\varphi_c = \frac{f_c}{f_c} \equiv 1 \ , \ \varphi_t = \frac{f_t}{f_c} \ , \ \varphi_{cc} = \frac{f_{cc}}{f_c} \tag{6}$$

are determined by $f_c$, $f_t$, $f_{cc}$ which are the static strengths of concrete for uniaxial compression, uniaxial tension, and biaxial compression taken from experimental data.

The material constants $a$, $b$, $c$ were determined based on experimental static investigations for uniaxial compression $\left\{(\sigma_{11} = \varphi_c \ , \ \sigma_{22} = \sigma_{33} = 0); \ \sigma_0 = \frac{1}{3}\varphi_c \ ; \ \tau_0 = \frac{\sqrt{2}}{3}\varphi_c \ ; \ \varphi = 0 \ ; \ \rho = \rho_c = 1\right\}$ and biaxial compression $\left\{(\sigma_{11} = \sigma_{22} = \varphi_{cc} \ , \ \sigma_{33} = 0); \ \sigma_0 = \frac{2}{3}\varphi_{cc} \ ; \ \tau_0 = \frac{\sqrt{2}}{3}\varphi_{cc} \ ; \ \varphi = \frac{\pi}{3} \ ; \ \rho = \rho_{cc} = \lambda\right\}$, and for uniaxial tension $\left\{(\sigma_{11} = \sigma_{22} = 0 \ , \ \sigma_{33} = -\varphi_t); \ \sigma_0 = -\frac{1}{3}\varphi_t \ ; \ \tau_0 = \frac{\sqrt{2}}{3}\varphi_t \ ; \ \varphi = \frac{\pi}{3} \ ; \ \rho = \rho_t = \lambda\right\}$, in the following forms:

$$a = \frac{\sqrt{2}}{6} \frac{\left[1-\left(\frac{\varphi_t}{\lambda}\right)^2\right](2\varphi_{cc}-1)-\left[\left(\frac{\varphi_{cc}}{\lambda}\right)^2-1\right](\varphi_t+1)}{\left(\frac{\varphi_{cc}}{\lambda}-1\right)(\varphi_t+1)-\left(1-\frac{\varphi_t}{\lambda}\right)(2\varphi_{cc}-1)}, b = \frac{\frac{2}{3}\left[1-\left(\frac{\varphi_t}{\lambda}\right)^2\right]+2\sqrt{2}\left(1-\frac{\varphi_t}{\lambda}\right)a}{\varphi_t+1}, c = \left(\frac{\sqrt{2}}{3}+a\right)^2 - \frac{1}{3}b \qquad (7)$$

In Figure 1, qualitative characteristics of the shape of the assumed limit surface equation for different cross-sections are shown.

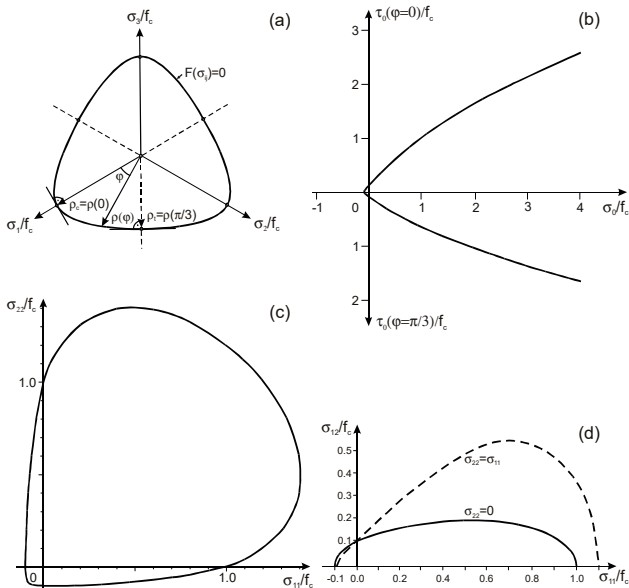

**Figure 1.** Limit surface for $K = 1$, $\varphi_t = 0.1$, $\varphi_{cc} = 1.1$: (**a**) section through the octahedral plane; (**b**) section through the Rendulic plane; compression curve for $\varphi = 0$, tension curve for $\varphi = \frac{\pi}{3}$; (**c**) plane stress state; (**d**) reduced plane stress state.

Comparative analysis carried by Stolarski [40] indicates good agreement of the assumed limit surface according to Equation (1) with other theoretical propositions (e.g., Podgórski [8], Ottosen [45], Klisiński [7], Willam and Warnke [24], Walther [46]) as well as with experimental results (e.g., Kupfer et al. [1], Mills and Zimmerman [2], Schickert and Winkler [4], Tasuji et al. [5], Luanay and Gachon [3]).

Proposed Equation (1) is described by a relatively inconsiderable number of material constants depending on the basic strength of concrete only. Moreover, introducing the evolution parameter $K$ enables the description of homothetic and isotropic expansion (for $K > 1$) or shrinkage (for $K < 1$) of the yield surface in the stress space.

*2.3. Dynamic Strength Criterion*

The basis of the model is the method of determination of the initial dynamic yield surface. For this purpose, the integral criterion of dynamic strength for concrete in complex stress states, is applied in the form proposed by Stolarski [40]:

$$\int_0^{t_d} [\psi(t)]^{\alpha_c} dt = t_{c0} \qquad (8)$$

where

$$\psi(t) = \frac{\sigma_{ij}(t)}{\sigma_{ij}^0} = \frac{\sigma_{int}(t)}{\sigma_{int}^0} \qquad (9)$$

is the proportionality function of any unrestricted elastic stressing process $\sigma_{ij}^t = \sigma_{ij}(t)$ to stress state $\sigma_{ij}^0$ satisfying the equation of static limit surface $F\left(\sigma_{ij}^0, K = 1\right) = 0$, expressing also the proportionality

of the stress intensity (invariant of stress deviator) $\sigma_{int}(t) = \sigma_{int}\left(\sigma_{ij}^t\right) = \sqrt{3J_2}$ corresponding with the dynamic stress state function $\sigma_{ij}^t$ to stress intensity $\sigma_{int}^0$ corresponding with stress state $\sigma_{ij}^0$, satisfying the equation of static limit surface.

Using the assumptions from Equation (9), the proportionality function $\psi(t)$ is determined based on the static yield surface according to Equation (1) for $K = 1$:

$$\psi(t) = \frac{2\left[\frac{\tau_0}{\rho(\varphi)}\right]^2}{2a\frac{\tau_0}{\rho(\varphi)} - b\sigma_0}\left\{\sqrt{1 - \frac{4(a^2 - c)\left[\frac{\tau_0}{\rho(\varphi)}\right]^2}{\left[2a\frac{\tau_0}{\rho(\varphi)} - b\sigma_0\right]^2}} - 1\right\}^{-1} \tag{10}$$

Material constants $\alpha_c$ and $t_{c0}$ were determined on the base of approximation of dynamic experimental results for uniaxial compression of concrete, and according to Stolarski [40], have the following values:

$$\alpha_c = 17.75 \, , \; t_{c0} = 0.180 \, s \tag{11}$$

Symbol $\psi(t)$ denotes:

$$\psi(t) = \begin{cases} \psi(t) & for \quad t \leq t_* \quad or \quad \psi(t) \geq 1 \\ 0 & for \quad t > t_* \quad if \quad \psi(t) < 1 \end{cases} \tag{12}$$

and $t_*$ is the time of attaining the static yield surface during the first stressing cycle, as shown in Figure 2a.

Time instant $t_d$ of attaining the dynamic limit surface $F\left(\sigma_{ij}^{t_d}, K = \psi_d\right) = 0$ is determined by the stress state $\sigma_{ij}^{t_d} = \sigma_{ij}(t_d)$ satisfying the dynamic strength criterion for concrete according to Equation (8):

$$\psi_d = \psi(t_d) \geq 1 \tag{13}$$

In Figure 2a,b, the interpretation of the dynamic strength criterion integration method is shown, both as the scheme of attaining the initial dynamic yield surface (Figure 2b), and during the variable stressing process (Figure 2a).

Determination of the dynamic strength coefficient (13) allows its use for the assumption of identical dynamic strength hardening for basic stress states:

$$\psi_d = \psi_c^d = \psi_t^d = \psi_{cc}^d \, , \; \psi_c^d = \frac{f_c^d}{f_c} \, , \; \psi_t^d = \frac{f_t^d}{f_t} \, , \; \psi_{cc}^d = \frac{f_{cc}^d}{f_{cc}} \tag{14}$$

where $f_c^d$, $f_t^d$, $f_{cc}^d$ are the dynamic strengths of concrete for uniaxial compression, uniaxial tension, and biaxial compression, and further extension of this assumption into any stress state $\psi_d = \frac{f_{ij}^d}{f_{ij}}$.

Thus, the dynamic strength coefficient $\psi_d$ can be treated as the scaling parameter defining basic dynamic strengths of concrete:

$$\varphi_c^d = \psi_d\varphi_c \equiv \psi_d \, , \; \varphi_t^d = \psi_d\varphi_t \, , \; \varphi_{cc}^d = \psi_d\varphi_{cc} \tag{15}$$

The proposed criterion of dynamic strength is used as the basic component of non-classical constitutive model of concrete. In this model, the initial elastic stage of dynamic deformation is limited by the time $t_d$ of attaining the dynamic strength coefficient of concrete $\psi_d$. This dynamic strength coefficient is treated in the elastic–plastic model of concrete as the starting and constant parameter in constitutive equations describing the initial condition of the evolution law for the yield surface in the stress space.

### 2.4. Plastic Potential Function

Plastic potential function is assumed in the form of a modified yield surface function:

$$G(\sigma_{ij}, K) = \left[ \frac{\tau_0}{\rho(\varphi)} + aK \right]^2 - \frac{b}{\beta} K \sigma_0 - cK^2 = 0 \tag{16}$$

where $\beta$ is the material constant adopted on the basis of appropriate comparative analysis with experimental results allowing to estimate the volume changes of material during plastic deformation.

Value of $\beta$ describes the angle between the direction of the plastic deformation vector compatible with the normal direction to the plastic potential surface $G(\sigma_{ij}, K)$ and the normal direction to the yield surface $F(\sigma_{ij}, K)$. This angle is responsible for changes in the volume of the material. The case of $\beta = 1$ means that the plastic potential function determines the associated plastic flow law $G(\sigma_{ij}, K) = F(\sigma_{ij}, K)$. In turn, $\beta > 1$ means that the plastic potential function is determined by the non-associated plastic flow rule $G(\sigma_{ij}, K) \neq F(\sigma_{ij}, K)$.

### 2.5. Strain Rate Decomposition

The assumption of strain rate decomposition into elastic $\dot{\varepsilon}_{ij}^e$ and plastic $\dot{\varepsilon}_{ij}^p$ parts is taken for the elastic–plastic model of concrete in the form:

$$\dot{\varepsilon}_{ij} = \dot{\varepsilon}_{ij}^e + \dot{\varepsilon}_{ij}^p \tag{17}$$

### 2.6. Non-Associated Flow Rule

Tensor of the plastic strain increment is defined by postulating the non-associated flow rule:

$$\dot{\varepsilon}_{ij}^p = \dot{\Lambda} \frac{\partial G(\sigma_{ij}, K)}{\partial \sigma_{ij}} \tag{18}$$

where $\dot{\Lambda}$ is the loading parameter.

### 2.7. Evolution Parameter

Parameter $K$ has the meaning of the scaling parameter for the basic strength of concrete and, on the other hand, can be interpreted as the evolution parameter for the dynamic yield surface in the stress space.

The following relation of the evolution parameter is introduced:

$$K = \begin{cases} 1 & & t < t_d \\ \psi_d & \varepsilon_{eff}^p \leq \varepsilon_f^p & t \geq t_d \\ \psi_d + \int_{t_f}^t \dot{K}(\tau) d\tau & \varepsilon_{eff}^p > \varepsilon_f^p & t > t_d \\ K_m & \varepsilon_{eff}^p = \varepsilon_u^p & t > t_d \\ 0 & \varepsilon_{eff}^p > \varepsilon_u^p & t > t_d \end{cases} \tag{19}$$

where $\varepsilon_{eff}^p$ is the effective plastic strain, $\varepsilon_f^p$ is the limit plastic strain in the perfect plastic flow phase, $\varepsilon_u^p$ is the limit plastic strain in the material softening phase, $t_f = t\left(\varepsilon_f^p\right)$ is the end instant of the perfectly plastic flow and commencement of the material softening phase, $K_m$ is the minimum value of the evolution parameter determining the end of the material softening phase and initiation of the failure phase: cracking (for tension) or crushing (for compression) of concrete.

The applied definition of the evolution parameter $K$ describes the four-phase idealization of concrete behavior. The following deformation phases are distinguished in this idealization: (1) elastic state until attaining the initial yield surface; (2) perfectly plastic flow in limited range of deformation; (3) material softening modeled as plastic flow on the transient yield surface, the isotropic shrinkage process which is controlled by variation $\dot{K}$ of the evolution parameter, dependent on the effective plastic strain rate and softening modulus, and modified in relation to the transient stress state; (4) residual (stress-free) state.

In Figure 2c,d, the interpretation is presented of the evolution parameter corresponding to the four-phase approximation of the concrete behavior for an example of uniaxial compression dependent on global strains (Figure 2c) and plastic effective strains (Figure 2d).

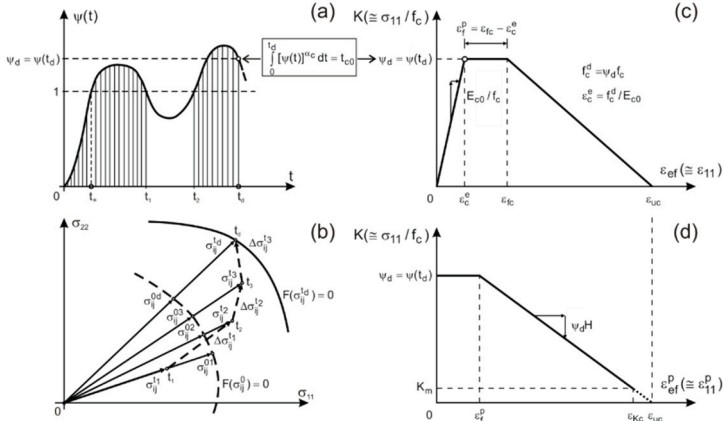

**Figure 2.** Evolution parameter: (**a**) dynamic strength criterion integration method during variable stressing process; (**b**) scheme of attaining the initial dynamic yield surface; (**c**) variation of evolution parameter for uniaxial compression depending on global strains; (**d**) variation of evolution parameter for uniaxial compression depending on effective plastic strains.

The following relation describing the variation $\dot{K}$ of the evolution parameter is introduced:

$$\dot{K} = \begin{cases} 0 & if \quad \varepsilon_{eff}^{p} \leq \varepsilon_{f}^{p} \\ \psi_d H\big(\sigma_{int}^{0}\big)\dot{\varepsilon}_{eff}^{p} & if \quad \varepsilon_{eff}^{p} > \varepsilon_{f}^{p} \\ 0 & if \quad \varepsilon_{eff}^{p} > \varepsilon_{u}^{p} \end{cases} \tag{20}$$

where $\dot{\varepsilon}_{eff}^{p}$ is the effective plastic strain rate; $H\big(\sigma_{int}^{0}\big)$ is the softening modulus.

The non-dimensional softening modulus $H(\sigma_{int}^{0})$ has the form:

$$H\big(\sigma_{int}^{0}\big) = -\frac{1}{\sigma_{int}^{0}\big(\varepsilon_{uc} - \varepsilon_{fc} + \varepsilon_{c}^{e}\big)} \, , \quad \varepsilon_{c}^{e} = \frac{f_{c}^{d}}{E_{c0}} \, , \quad f_{c}^{d} = \psi_d f_c \tag{21}$$

where $\varepsilon_{fc}$ and $\varepsilon_{uc}$ are the global limit strains for the perfectly plastic flow phase and for the material softening phase; $E_{c0}$ is the initial modulus of elasticity.

Limit plastic strains $\varepsilon_{f}^{p}$ and $\varepsilon_{u}^{p}$ for the perfectly plastic flow phase and for the material softening phase are taken in the forms

$$\varepsilon_{f}^{p} = \sigma_{int}^{0d}\big(\varepsilon_{fc} - \varepsilon_{c}^{e}\big) \, , \quad \varepsilon_{u}^{p} = \sigma_{int}^{0d}\big(\varepsilon_{Kc} - \varepsilon_{c}^{e}\big) \, , \quad \varepsilon_{Kc} = \varepsilon_{uc} - \frac{K_m}{\psi_d}\big(\varepsilon_{uc} - \varepsilon_{fc}\big) \tag{22}$$

where $\sigma_{int}^{0d} = \sigma_{int}^{0}(t_d) = \frac{\sigma_{int}(t_d)}{\psi(t_d)}$ is determined for the instant $t = t_d$ of dynamic strength criterion fulfillment in an analogical way as in Equation (9).

Global limit strains $\varepsilon_{fc}$ and $\varepsilon_{uc}$ are taken on the base of the analysis of dynamic experimental results for uniaxial compression of concrete (Bazenov [12], Rostasy and Hartwich [14], Dilger et al. [17], Kowalczyk and Dilger [18]), independently of strain rate:

$$\varepsilon_{fc} = 0.002 \, , \; \varepsilon_{uc} = 0.006 - 0.012 \tag{23}$$

where lower values of $\varepsilon_{uc}$ might be used for high-class concrete and greater ones for low- and mean-class concrete.

Effective plastic strain $\varepsilon_{eff}^{p}$ is defined as follows:

$$\varepsilon_{eff}^{p}(t) = \int_{t_d}^{t} \dot{\varepsilon}_{eff}^{p}(\tau)d\tau \tag{24}$$

In turn, effective plastic strain rate $\dot{\varepsilon}_{eff}^{p}$ is assumed as the modified form of plastic strain rate intensity in which the influence of transverse strains is neglected:

$$\dot{\varepsilon}_{eff}^{p} = \dot{\varepsilon}_{int}^{p}(\nu_c = 0) = \sqrt{\frac{3}{2}} \sqrt{\dot{e}_{ij}^{p}(\nu_c = 0)\dot{e}_{ij}^{p}(\nu_c = 0)} \tag{25}$$

where $\dot{e}_{ij}^{p}$ are the components of plastic strain rate deviator.

## 2.8. Degradation of Modulus of Elasticity

The degradation of elasticity modulus is taken into consideration. The following rule describes the modulus of elasticity:

$$E_c = E_{c0}(1 - d) \tag{26}$$

Degradation parameter has the form:

$$d = 1 - k^{\alpha_e} \tag{27}$$

where function $k$ depends on the evolution parameter (Equation (19)) only for compressive processes of deformation in the following way:

$$k = \begin{cases} 1 & if \quad t < t_d \\ K_c/\psi_d & if \quad t \geq t_d \end{cases} , \quad K_c = K \quad for \quad \sigma_0 \geq \sigma_0^* \tag{28}$$

and $\alpha_e$ is the material constant.

The established value of the mean normal stress $\sigma_0^*$ separates fully compressive processes of deformation and it is assumed as $\sigma_0^* = 0.25 \cdot K \cdot f_c$.

Figure 3 illustrates the degradation of the modulus of elasticity for the example of uniaxial cyclic compression for different values of the material constant $\alpha_e$ on the background of the experimental results presented by Sinha et al. [20].

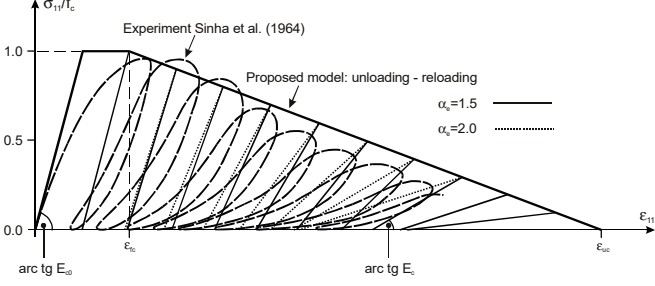

**Figure 3.** Degradation of modulus of elasticity.

## 2.9. Incremental Constitutive Relations

Linear elastic material properties are described by generalized Hooke's law, which, considering the strain rates decomposition and assumed non-associated flow rule, can be written in the form:

$$\dot{\sigma}_{ij} = C_{ijkl}\dot{\varepsilon}_{kl}^e = C_{ijkl}\left(\dot{\varepsilon}_{kl} - \dot{\varepsilon}_{kl}^p\right) = C_{ijkl}\left(\dot{\varepsilon}_{kl} - \dot{\Lambda}\frac{\partial G}{\partial \sigma_{kl}}\right), \tag{29}$$

where $C_{ijkl} = \lambda_c \delta_{ij}\delta_{kl} + \mu_c\left(\delta_{ik}\delta_{jl} + \delta_{il}\delta_{jk}\right)$ is the tensor of elastic constants; $2\mu_c = \frac{E_c}{1+\nu_c}$ is the shear modulus; $3\kappa_c = \frac{E_c}{1-2\nu_c}$ is the bulk modulus; $3\lambda_c = 3\kappa_c - 2\mu_c$ is the Lamé constant; $\nu_c$ is the transverse strains coefficient; $E_c$ is the deformation modulus according to (26), treated as a step function, variable step by step, but constant in each interval between steps.

## 2.10. Elastic–Plastic Loading Process

Elastic–plastic flow takes place if the stress state satisfies the following conditions:

$$F\left(\sigma_{ij}, K\right) = 0, \; \dot{F}\left(\sigma_{ij}, K\right) = 0 \tag{30}$$

Consistency condition (Equation (30)$_2$), defining the active plastic flow processes:

$$\dot{F}\left(\sigma_{ij}, K\right) = \frac{\partial F}{\sigma_{ij}}\dot{\sigma}_{ij} + \frac{\partial F}{\partial K}\dot{K} = 0 \tag{31}$$

considering variation $\dot{K}$ of the evolution parameter (Equation (20)$_2$):

$$\dot{K} = \psi_d H\left(\sigma_{int}^0\right)\dot{\varepsilon}_{eff}^p = -\dot{\Lambda}M\left(\psi_d, H, m_{eff}\right) \tag{32$_1$}$$

$$M\left(\psi_d, H, m_{eff}\right) = -\psi_d H\left(\sigma_{int}^0\right)m_{eff}\left(\frac{\partial G}{\partial \sigma_{ij}}\right) \tag{32$_2$}$$

$$m_{eff} = \sqrt{\frac{3}{2}}\sqrt{m_{ij}m_{ij}}, \; m_{ij} = \frac{\partial G}{\partial \sigma_{ij}} \tag{32$_{3,4}$}$$

enables the determination of the loading parameter on the base of the relation:

$$\dot{\Lambda} = \frac{\frac{\partial F}{\partial \sigma_{ij}}C_{ijkl}\dot{\varepsilon}_{kl}}{\frac{\partial F}{\partial K}M\left(\psi_d, H, m_{eff}\right) + \frac{\partial F}{\partial \sigma_{ij}}C_{ijkl}\frac{\partial G}{\partial \sigma_{kl}}} \tag{33}$$

Loading and unloading processes are defined in the following way:

$$\begin{array}{llllll}
loading: & \dot{\Lambda} > 0 & if & F = 0 & and & \dot{F} = 0 \\
unloading: & \dot{\Lambda} = 0 & if & F < 0 & or & F = 0 \; and \; \dot{F} < 0
\end{array} \tag{34}$$

For the perfectly plastic phase of deformation, the evolution parameter according to Equation (19)$_2$, determining the initial dynamic yield surface, has the constant value $K = \psi_d$, thus $\dot{K} = 0$ and $M\left(\psi_d, H, m_{eff}\right) = 0$.

In turn, for the material softening phase of deformation, parameter $K$ according to Equation (19)$_3$ determines instantaneous yield surface $F\left(\sigma_{ij}, K\right) = 0$. The active plastic flow state determined by the loading parameter $\dot{\Lambda} > 0$, takes place only in the shrinkage process of the yield surface because variation of the evolution parameter is negative, $\dot{K} < 0$.

Since the so-called trial stress rate in the material softening phase of deformation is realized as elastic stress rate-dependent on the total strain rate:

$$\dot{\sigma}_{ij}^e = C_{ijkl}\dot{\varepsilon}_{kl} \tag{35}$$

thus, distinction of loading and unloading processes can be interpreted in the following way:

$$loading: \ \frac{\partial F}{\partial \sigma_{ij}}\dot{\sigma}_{ij}^e > 0 \, , \ loading: \ \frac{\partial F}{\partial \sigma_{ij}}\dot{\sigma}_{ij}^e \le 0 \tag{36}$$

under an additional condition:

$$\frac{\partial F}{\partial K}M\big(\psi_d, H, m_{eff}\big) + \frac{\partial F}{\partial \sigma_{ij}}C_{ijkl}\frac{\partial G}{\partial \sigma_{kl}} > 0 \tag{37}$$

The foregoing conditions of loading and unloading processes result also from analysis of the Il'Yushin [43] postulate of positive plastic power $W^p = \sigma_{ij}\dot{\varepsilon}_{ij}^p > 0$, which is a fundamental condition for the description of plastic flow of materials demonstrating the material softening effect (Ohtani and Chen [47]).

### 2.11. Cracking and Crushing Mechanism

The applied deformation model of concrete enables the simplified modeling of the failure mechanism. This mechanism results from the applied softening rule, which assumes the gradual loss of material load-capacity until attaining the residual stress state during the tension or compression processes.

In Figure 4, basic cases of loading and unloading paths in the deformation processes of the assumed model of concrete are schematically shown.

Three different states of the failure mechanism can be defined at the moment of attainment of the minimal (residual) value of the evolution parameter:

$$\left.\begin{array}{ll} K_{mt} = K_m & \left\{\begin{array}{lll} if & \sigma_0 < 0 & cracking\ state \\ if & 0 < \sigma_0 < \sigma_0^* & semi-cracking\ state \\ \end{array}\right. \\ K_{mc} = K_{mt} = K_m & if \quad \sigma_0 \ge \sigma_0^* \quad crushing\ state \end{array}\right\} \tag{38}$$

The cracking state can be attained in the monotonic tension process (Figure 4a) or in the cyclic, convertible unloading from the compression process that reverses the loading in the tension process (Figure 4b,c). The cracking state does not reduce the compression strength, and the reloading process in compression is possible after closing the generalized, volumetric crack.

The crack opening and closing mechanisms are determined by the following conditions:

$$\left.\begin{array}{l} \varepsilon_0 < \varepsilon_0^* - crack\ opening \\ \varepsilon_0 \ge \varepsilon_0^* - crack\ closing \end{array}\right\} \tag{39}$$

where $\varepsilon_0^*$ is the last converged limit volumetric strain corresponding to the instant of attainment of the residual stress state in the tension process.

The semi-failure state, being analogical to the cracking state, is characterized by its capability to the re-compressive process if the current volumetric deformation is greater than the limit volumetric strain $\varepsilon_0 \ge \varepsilon_0^*$, attained at the instant of the semi-failure state of concrete in the previous cycle of deformation (Figure 4e,f).

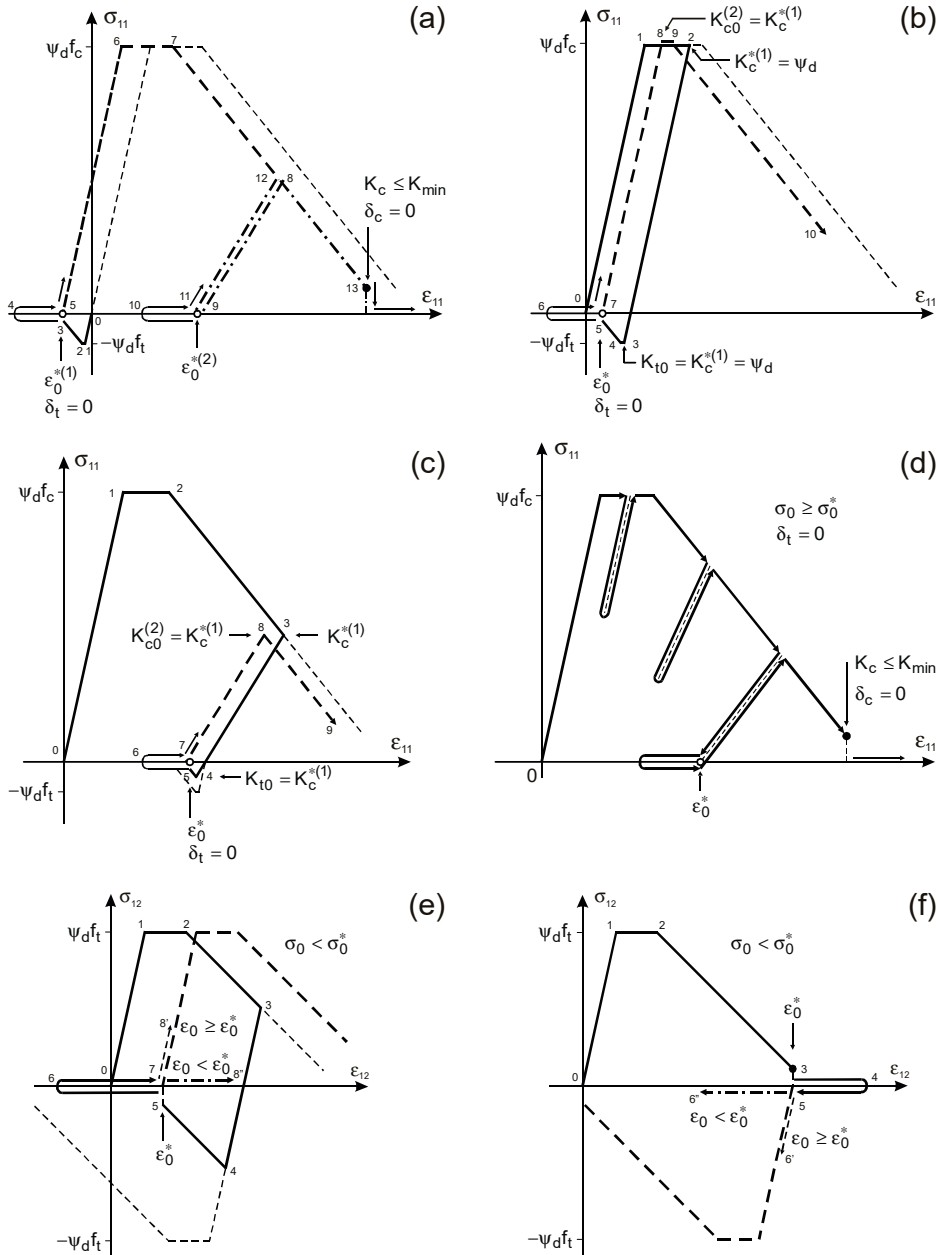

**Figure 4.** Schemes of approximate modeling of cracking and crushing for concrete (description in text): (**a**) crack opening in tension path 0-1-2-3-4 / re-compression after crack closing path 5-6-7-8 / unloading path 8-9 / secondary crack opening path 9-10 / secondary crack closing and re-compression path 11-12-13; (**b**) compression path 0-1-2 / unloading and tension path 2-3-4-5 / crack opening path 5-6 / secondary loading and crack closing path 6-7 / re-compression path 7-8-9-10; (**c**) advanced compression to material softening range path 0-1-2-3 / unloading with current deformation modulus and tension path 3-4-5 / crack opening path 5-6 / secondary loading and crack closing path 6-7 / re-compression path 7-8-9; (**d**) attaining of the crushing state in the material softening process with local unloading and re-loading paths; (**e**) advanced shear stress state: loading path 0-1-2-3 / unloading path 3-4-5 / loss of load capacity with associated tension path 5-6 / re-loading path 6-7 with associated compression or tension; (**f**) attaining of the semi-failure state in the shear stress state path 0-1-2-3 / loss of load capacity 3-4 / re-loading path 4-5 with associated compression or tension.

The crushing state determines the total loss of stress carrying capacity of the concrete (Figure 4d).

The assumed interpretation of the cracking and crushing mechanism can be described by means of the indicator defined as the unitary, active cross-section of concrete:

$$\delta = \left\{ \begin{array}{ll} \delta_c & if \quad \sigma_0 \geq 0 \\ \delta_c \delta_t & if \quad \sigma_0 < 0 \end{array} \right. \tag{40}$$

where $\delta_c = 1$ and $\delta_t = 1$ are the initial values of the partial indicators that change their values only once, just at the moment when the residual stress states are attained

$$\begin{array}{ll} \delta_c = 0 & if \quad K_{mc} = K_m \\ \delta_t = 0 & if \quad K_{mt} = K_m \end{array} \tag{41}$$

Assuming that indicator $\delta$ determines the evolution parameter and the stress state in concrete as follows: $\widetilde{K} = \delta \cdot K$ and $\widetilde{\sigma}_{ij} = \delta \cdot \sigma_{ij}$, the residual (stress-free) state $\widetilde{\sigma}_{ij} = 0$, is determined directly by the indicator value $\delta = 0$ attained in the crushing process of the concrete $\delta_c = 0$ or in the cracking (or semi-failure) process $\delta_t = 0$.

## 3. Comparative Analysis

### 3.1. Assumptions for Analysis

Hereafter is presented a comparison of the proposed model of deformation of concrete with static and dynamic experimental curves $\sigma = \sigma(\varepsilon)$ available in the literature and known theoretical models of dynamic deformation of concrete. The comparisons relate to uniaxial and biaxial stress state with respect to static curves and uniaxial state of stress with respect to dynamic curves.

In each case, the proposed model is defined by material data contained in the works used for comparison (i.e., concrete strength for uniaxial compression $f_c$, coefficients $\varphi_t$, $\varphi_{cc}$ according to Equation (6) defining the concrete strength for uniaxial tension and biaxial compression, initial modulus of elasticity $E_{c0}$, and factor of transverse strain $\nu_c$).

Moreover, material constants occurring in the criterion of dynamic strength (Equation (8)) $\alpha_c = 17.75$, $t_{c0} = 0.180\ s$ according to Equation (11) and in other material functions were adopted: coefficient $K_m = 0.1$ in a function of changes of the softening parameter (Equation (19)); basic limit strains $\varepsilon_{fc}$, $\varepsilon_{uc}$ with values according to Equation (23) or another, adapted to the considered comparative results; material constant $\beta$ defining the plastic potential function (Equation (16)).

### 3.2. Comparisons for Static Tests

A comparison of the proposed model with the experimental stress–strain curves was carried out on the basis of the experimental results of Kupfer et al. [1].

In Figure 5 for uniaxial compression ($\sigma_{11} = \sigma$, $\sigma_{22} = \sigma_{33} = 0$), theoretical results were marked with solid lines, while the experimental curves according to [1] were marked with dashed lines. In Figure 5a, the variation of stress as a function of strain $\varepsilon_{11}$ "forcing" the considered stress state and as a function of transverse strains $\varepsilon_{22} = \varepsilon_{33}$ resulting from volumetric strains $\varepsilon_0 = \frac{1}{3}\varepsilon_{kk}$, was presented for different $\beta$ values in the plastic potential function (Equation (16)). Graphs of stress changes as a function of strains $\varepsilon_{kk}$ are shown in Figure 5b. The result for $\beta = 1$ corresponds to the solution obtained for the associated flow rule $G = F$.

The variability of the presented stress–strain curves is characteristic for the introduced model. The occurrence of the linear elastic phase, the perfectly plastic flow phase in a limited range of strains, and the material softening phase is observed in Figure 5a. The change of volumetric strains in these phases is typical for brittle materials (Figure 5b). In the initial elastic phase, the volume strains increase and the material undergoes to compaction. In the perfectly plastic flow phase, the volume

strains decrease. In the softening phase the material undergoes loosening as well, but in this case, with decreasing stresses.

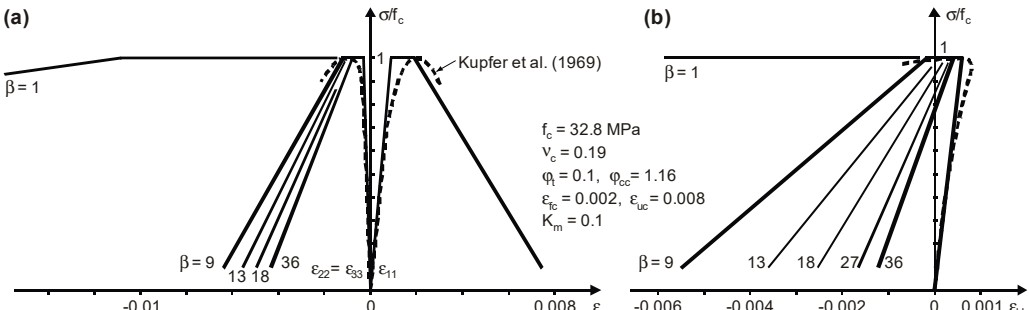

**Figure 5.** Comparison of the proposed model with experimental results for uniaxial compression: (**a**) graph of stress vs. longitudinal and lateral strains; (**b**) graph of stress vs. volumetric strains.

Figure 6 presents analogous results for the case of symmetrical, biaxial compression ($\sigma_{11} = \sigma_{22} = \sigma$, $\sigma_{33} = 0$). Figure 6a relates to stress changes $\sigma_{11} = \sigma_{22} = \sigma$ in the function of strains $\varepsilon_{11} = \varepsilon_{22}$ and in the function of transverse strains $\varepsilon_{33}$ determined from the law of volume changes. Figure 6b presents graphs of stress changes $\sigma_{11} = \sigma_{22} = \sigma$ in the function of volumetric strains $\varepsilon_0 = \frac{1}{3}\varepsilon_{kk}$.

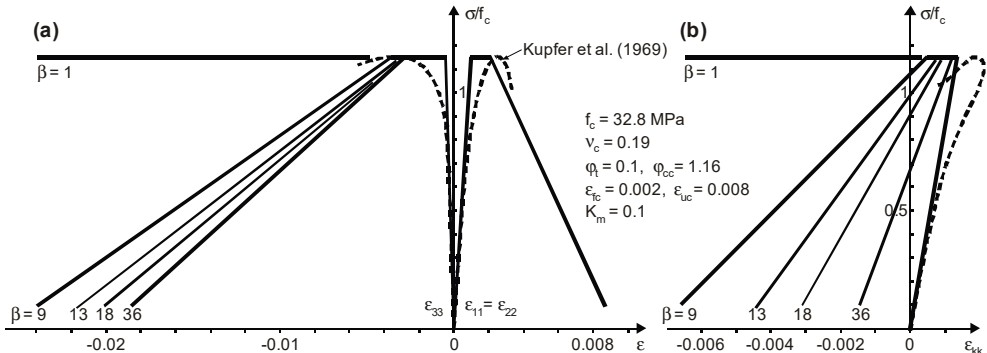

**Figure 6.** Comparison of the proposed model with experimental results for biaxial compression: (**a**) graph of stress vs. longitudinal and lateral strains; (**b**) graph of stress vs. volumetric strains.

On the basis of presented comparisons of theoretical solutions with experimental results, it can be concluded that the graphs of probable stress changes as a function of volumetric strains and as a function of transverse strains should be consistent with the graph of stress changes as a function of strains "forcing" the considered stress state. Such theoretical solutions—characterized by the presence of a distinct range of material softening in all of the mentioned relationships—are obtained for the constant $\beta \gg 1$. The solution for $\beta = 1$—corresponding to the associated flow rule—practically does not describe changes in the volume of material and can be considered as unrealistic. However, in the literature there are no experimental results concerning the changes in volumetric strains of the concrete over the whole range of material softening, it can be assumed that the theoretical solutions obtained for interval $\beta = \langle 9, \ 36 \rangle$ approximately describes the real behavior of the concrete.

Figure 7 presents the comparison of the proposed model with the theoretical solution developed on the basis of the degradation theory, placed in the paper by Dragon and Mróz [27]. The solution concerns a plane, symmetrical stress state ($\sigma_{11} = \sigma_{22} = \sigma$, $\sigma_{33} = 0$). Material data were adopted according to the cited work. The presented results were obtained for the constant $\beta$ occurring in the plastic potential function (Equation (16)) with the values in the interval $\beta = \langle 18, \ 27 \rangle$. The general agreement of the proposed model with the solution [27] is noteworthy, although both solutions are qualitatively different. This agreement relates primarily to the points of the maximum stresses shown in the graphs $\sigma = \sigma(\varepsilon)$. The level of maximum stresses determined as material strength and strain values corresponding to

this level of stresses are approximated by the proposed model as a two-phase, elastic–perfectly plastic behavior of the material. The third phase of material behavior is approximated in accordance with the adopted material softening law and is characterized—in contrary to the solution [27]—by practically constant increments of loosening the material as it is softened and reducing stresses (Figure 7b).

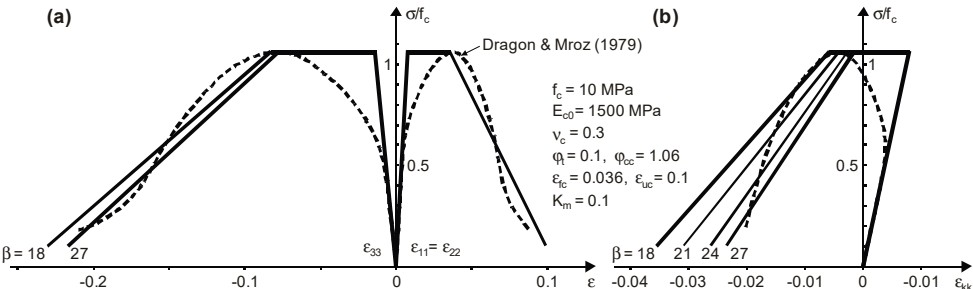

**Figure 7.** Comparison of the proposed model with Dragon's and Mróz's [27] solution for biaxial compression: (**a**) graph of stress vs. longitudinal and lateral strains; (**b**) graph of stress vs. volumetric strains.

### 3.3. Comparisons for Dynamic Tests

The first group of dynamic tests refers to verification of the proposed dynamic strength criterion with some experimental dynamic results available in the literature. These comparisons were performed for uniaxial compressive loading tests, biaxial compressive–compressive loading tests, and biaxial tensile–compressive loading tests.

Using the dynamic strength criterion of Equation (8) for the tests of input stress rate $\dot{\sigma} = const$, with material constants of Equation (11), enables the comparison of obtained theoretical low (C10—red line), medium (C50—green line), and high (C100—blue line) strength classes of concrete with experimental results for the uniaxial compression presented in the paper of Bischoff and Perry [48] (Figure 8). The general compatibility of the results obtained in a large range of strain rates confirms the validity of the proposed criterion for determining the dynamic strength of concrete.

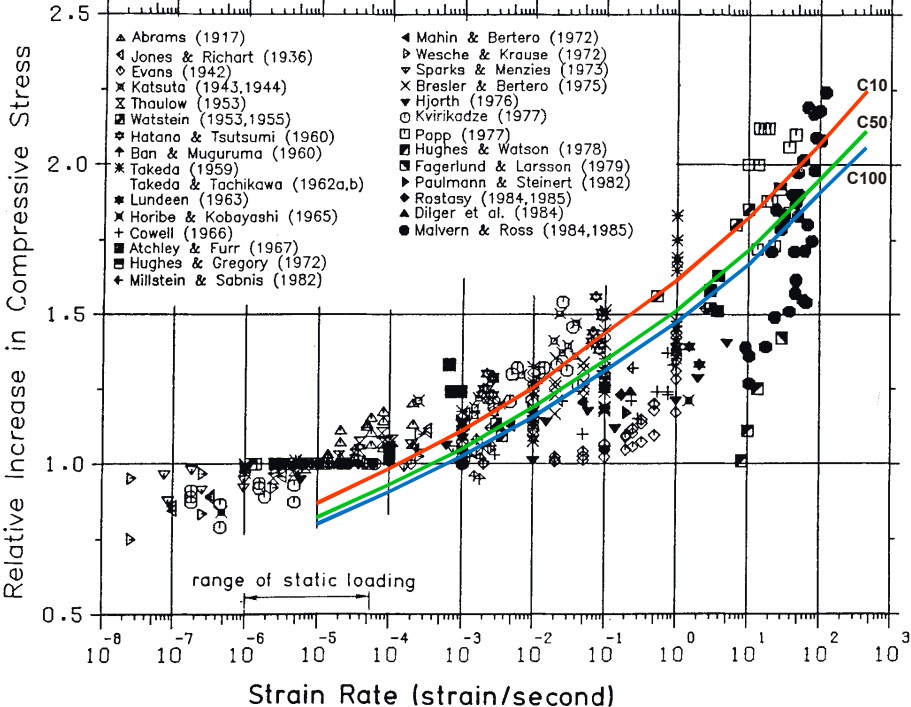

**Figure 8.** The results of the proposed dynamic strength criterion of concrete on the background of the results for the uniaxial compression presented by Bischoff and Perry [48].

Figure 9 presents the comparison of the proposed dynamic strength criterion with selected experimental results carried out by Yan and Lin [49] for biaxial loading tests with the type of compression–compression. The experimental results were conducted for concrete with uniaxial compressive strength $f_c = 9.84\ MPa$. The experimental results presented in Figure 9 relate to the loading process with constant ratio lateral stress $\sigma_{22}$ to axial stress $\sigma_{11}$ of values $\frac{\sigma_{22}}{\sigma_{11}} = \frac{0}{1}; \frac{0.25}{1}; \frac{0.5}{1}; \frac{0.75}{1}; \frac{1}{1}$. For quasi-static strain rate $\dot{\varepsilon} = 10^{-5}\ s^{-1}$ stresses have been obtained with values of $\frac{\sigma_{22}/f_c}{\sigma_{11}/f_c} = \frac{0}{1.0}; \frac{0.38}{1.51}; \frac{0.82}{1.64}; \frac{1.25}{1.67}; \frac{1.42}{1.42}$, while for strain rate $\dot{\varepsilon} = 10^{-2}\ s^{-1}$, the dynamic stresses have been obtained with values of $\frac{\sigma_{22}/f_c}{\sigma_{11}/f_c} = \frac{0}{1.25}; \frac{0.44}{1.74}; \frac{0.93}{1.85}; \frac{1.42}{1.90}; \frac{1.83}{1.83}$. The static limit curve of Equation (1) was determined for the test data including $\varphi_{cc} = 1.42$ and assuming $K = 1$, $\varphi_t = 0.1$. In turn, the dynamic limit curve of Equation (1) was determined for the dynamic strength coefficient $K = \psi_d = 1.27$ calculated according to criterion of Equation (8) with material constants from Equation (11). Comparison of the results indicates a good agreement between the proposed theoretical limit curves with the experimental results, especially in relation to uniaxial and symmetrically biaxial compression.

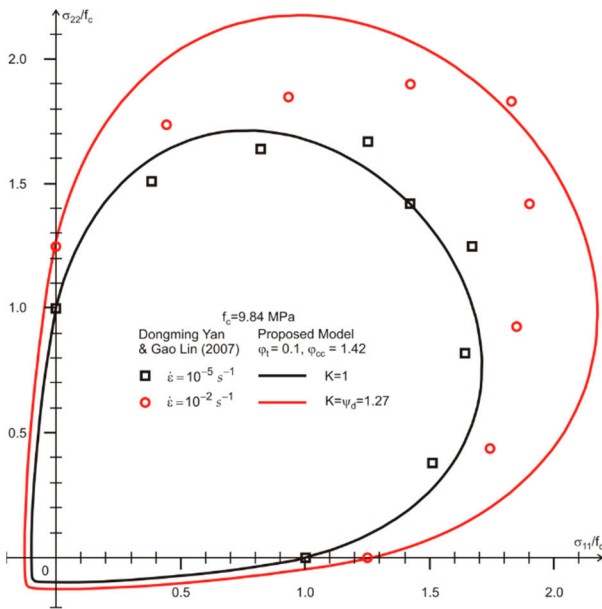

**Figure 9.** Comparison of the proposed dynamic strength criterion with experimental results by Yan and Lin [49] for biaxial compressive–compressive loading tests.

Comparison of the proposed dynamic strength criterion with selected experimental results conducted by Ping and Peng [50] for biaxial compressive–compressive loading tests, is presented in Figure 10. The experimental results were conducted for concrete with compressive strength $f_c = 20.1\ MPa$. The experimental results presented in Figure 10 relate to the process of loading with the constant lateral confining pressure of values $\sigma_{22} = (0\ ;4\ ;7\ ;10)\ MPa$ (i.e., $\frac{\sigma_{22}}{f_c} = 0\ ;0.20\ ;0.35\ ;0.50$). For quasi-static strain rate $\dot{\varepsilon} = 10^{-5}\ s^{-1}$ axial stresses have been obtained with values of $\frac{\sigma_{11}}{f_c} = 1.00\ ;1.09\ ;1.17\ ;1.25$. For strain rate $\dot{\varepsilon} = 10^{-2}\ s^{-1}$ dynamic axial stresses have been obtained with values of $\frac{\sigma_{11}}{f_c} = 1.36\ ;1.39\ ;1.46\ ;1.60$. The static limit curve of Equation (1) was determined for the test data and assuming $K = 1$, $\varphi_t = 0.1$, $\varphi_{cc} = 1.1$. The dynamic limit curve of Equation (1) was determined for the dynamic strength coefficient $K = \psi_d = 1.23$ calculated according to criterion of Equation (8) with material constants from Equation (11). Comparison of the results indicates a good agreement between the proposed theoretical limit curves and the experimental results.

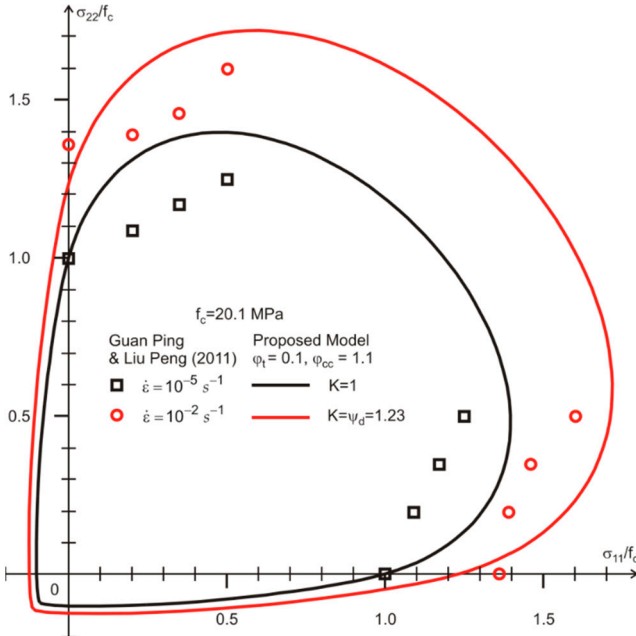

**Figure 10.** Comparison of the proposed dynamic strength criterion with experimental results by Ping and Peng [50] for biaxial compressive–compressive loading tests.

In Figure 11, comparison of the proposed dynamic strength criterion with selected experimental results conducted by Shiming and Yupu [51] for biaxial tensile–compressive loading tests, is presented.

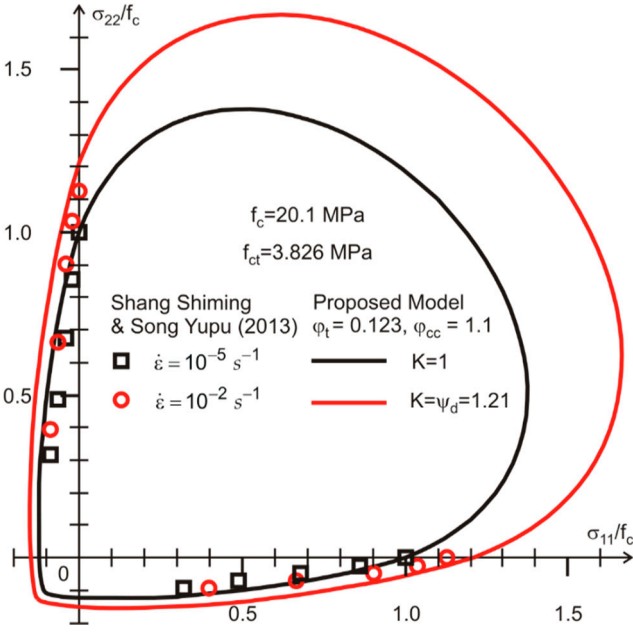

**Figure 11.** Comparison of the proposed dynamic strength criterion with experimental results by Shiming and Yupu [51] for biaxial tensile–compressive loading tests.

The experimental results were conducted concrete with uniaxial compressive $f_c = 31.2 \; MPa$ and tensile $f_{ct} = 3.826 \; MPa$ strengths. The experimental results presented in Figure 11 relate to the loading process with $\sigma_{33} = 0$ and constant values of tensile stresses $\sigma_{22} = (0 \, ; \, 0.7 \, ; 1.4 \, ; 2.1 \, ; 2.8) \; MPa$ (i.e., $\frac{\sigma_{22}}{f_c} = 0 \, ; -0.022 \, ; -0.045 \, ; -0.067 \, ; -0.090$). For strain rate $\dot{\varepsilon} = 10^{-5} \; s^{-1}$ (set as static loading rate) axial stresses have been obtained with values of $\frac{\sigma_{11}}{f_c} = 1.00 \, ; 0.857 \, ; 0.679 \, ; 0.487 \, ; 0.318$. For strain rate $\dot{\varepsilon} = 10^{-2} \; s^{-1}$

dynamic axial stresses have been obtained with values of $\frac{\sigma_{11}}{f_c} = 1.128$ ; $1.037$ ; $0.901$ ; $0.663$ ; $0.398$. The static limit curve of Equation (1) was determined for the test data including $\varphi_t = 0.123$ and assuming $K = 1$, $\varphi_{cc} = 1.1$. The dynamic limit curve of Equation (1) was determined for the dynamic strength coefficient $K = \psi_d = 1.21$ calculated according to criterion of Equation (8) with material constants from Equation (11). Comparison of the results indicates that the proposed theoretical limit curves are in quite good agreement with the experimental results.

The second group of dynamic tests refers to the comparison of the proposed model with experimental dynamic curves available in the literature and known theoretical models of dynamic deformation of concrete. These comparisons were performed for uniaxial compression ($\sigma_{11} = \sigma$, $\sigma_{22} = \sigma_{33} = 0$); the most reliable for the assessment of the dynamic load capacity of concrete.

Figure 12 shows the compatibility of the proposed model in terms of strains $\langle 0, \ \varepsilon_{fc} \rangle$ with some results of Watstein [10]. The experimental results obtained in dynamic and static tests, were marked with continuous and dotted lines, respectively, and the proposed idealization was marked with a discontinuous line. Noteworthy is the increasing linear range of deformation observed in the experimental dynamic curves $\sigma - \varepsilon$ with respect to the static curves. The proposed model approximates well the experimental results for the tested range of strains.

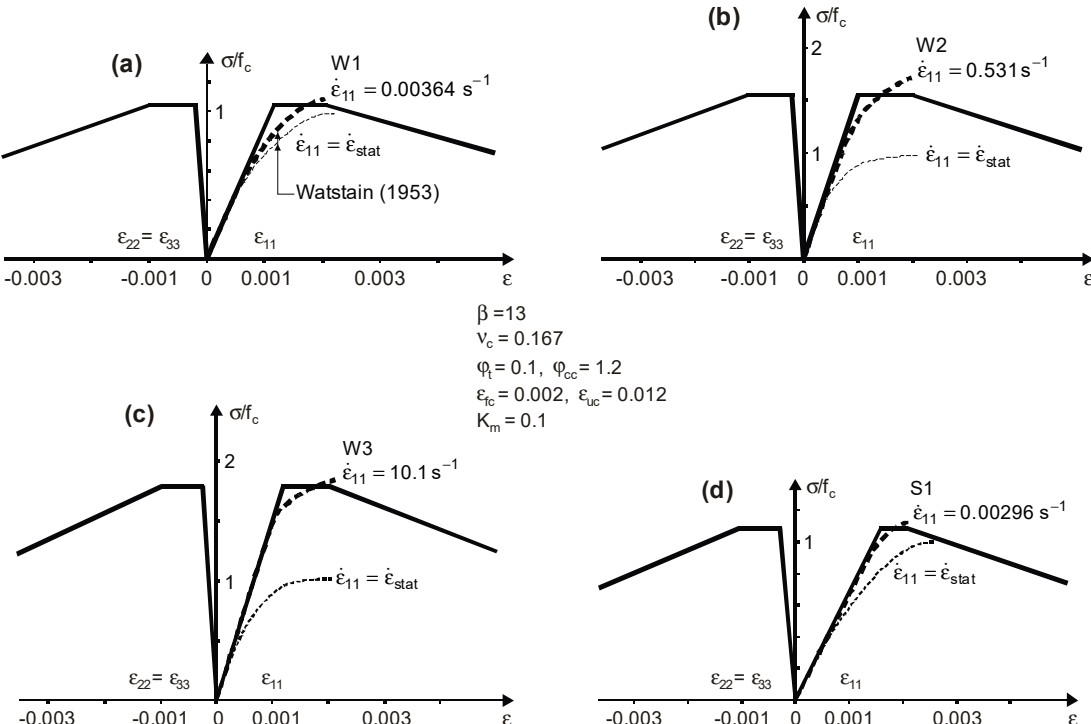

**Figure 12.** Comparison of the proposed model with Watstein's [10] dynamic experimental results for different test samples and strain rate: (**a**) W1 $\dot{\varepsilon} = 0.00364\ s^{-1}$; (**b**) W2 $\dot{\varepsilon} = 0.531\ s^{-1}$; (**c**) W3 $\dot{\varepsilon} = 10.1\ s^{-1}$; (**d**) S1 $\dot{\varepsilon} = 0.00296\ s^{-1}$.

An agreement in results for the same range of strains $\langle 0, \ \varepsilon_{fc} \rangle$ also demonstrate the comparisons presented in Figure 13, based on the work of Kowalczyk and Dilger [18]. The disagreement of the model with experimental results in the range of material softening is influenced by the length of the strain measuring base $\varepsilon^{(24)}$ which is equal to the length of the sample (i.e., 24" $\cong$ 0.60 $m$).

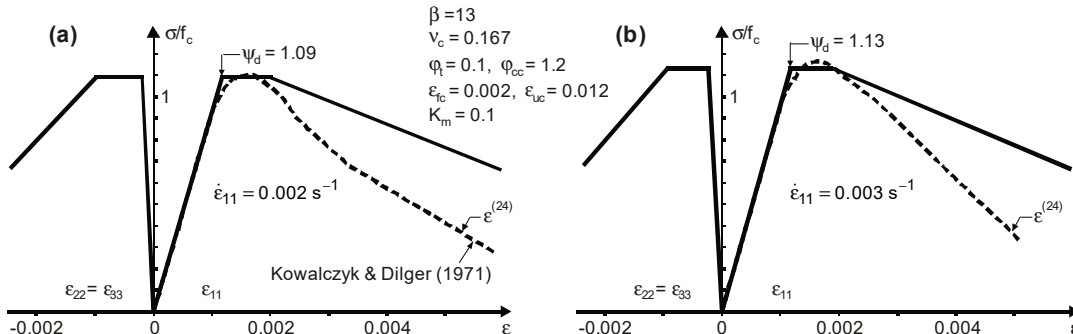

**Figure 13.** Comparison of the proposed model with Kowalczyk's and Dilger's [18] dynamic experimental results for different strain rate: (**a**) $\dot{\varepsilon} = 0.002\ s^{-1}$; (**b**) $\dot{\varepsilon} = 0.003\ s^{-1}$.

The authors of the work [18] consider that the authoritative basis for the softening range is the base equal to the lateral dimension of the sample (in this case $6'' \cong 0.15\ m$) corresponding to the length of the destruction zone in the middle part of the sample. For such a strain base $\varepsilon^{(6)}$, the comparative analysis for the partially normalized dynamic curve $\frac{\sigma_{11}}{f_c^d} - \varepsilon_{11}$ averaged from many samples was carried out in Figure 14. In this case, good agreement of the proposed model is observed also in the range of material softening. The descending branch of the $\frac{\sigma_{11}}{f_c^d} - \varepsilon_{11}$ curve is characterized by the value of strain $\varepsilon_k$ corresponding to the stress decrease by 15% (i.e., for $k = \frac{\sigma_{11}}{f_c^d} = 0.85$). For the assumed values of the limit strains $\varepsilon_{fc}$ and $\varepsilon_{uc}$, the obtained strain value $\varepsilon_k = 3.5\frac{o}{oo}$ is consistent with the experimental results presented in the paper [18].

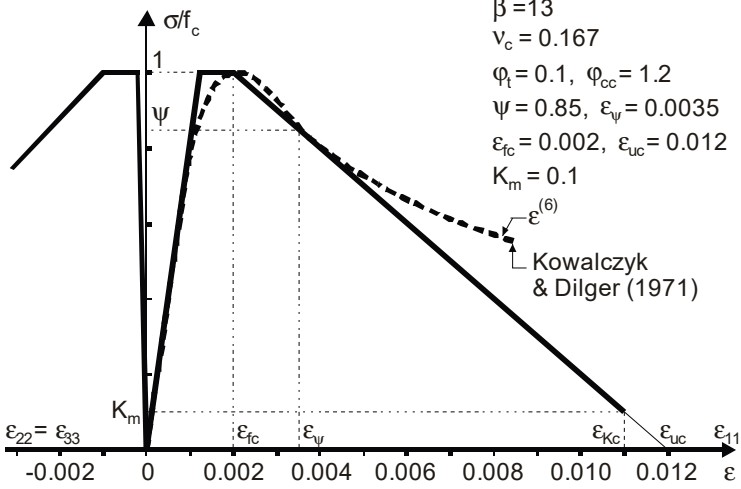

**Figure 14.** Comparison of the proposed model with the average dynamic curve according to Kowalczyk and Dilger [18].

In the following, the comparison of the proposed model with known models of dynamic concrete deformation is presented. The first of the presented comparisons concerns the model proposed by Nilsson [6]. In Figure 15, the curves obtained according to the Nilsson model are marked with thin lines and the bold lines show the proposed idealization. The individual results were determined for different strain rates being a multiple of $\dot{\varepsilon}_{11}^0 = 2 \times 10^{-6} s^{-1}$ value corresponding to the static test. Good agreement of the presented solutions is observed in relation to the level of stresses corresponding to dynamic strength.

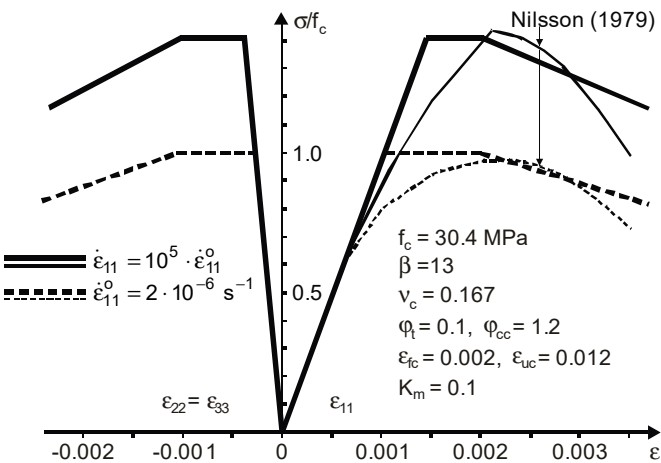

**Figure 15.** Comparison of the proposed model with Nilsson's [6] model.

Another theoretical model is presented by Soroushian at al. [15] for uniaxial compression at a constant strain rate $\dot{\varepsilon} = const$. In Figure 16, thin lines denote solutions obtained in [15], and bold lines denote a solution according to the proposed model. The presented comparisons refer to concrete with static strength $f_c = 14.3$ $MPa$. The individual results were obtained for multiples of the strain rate $\dot{\varepsilon}_{11}^0 = 10^{-5} s^{-1}$ corresponding to the static test. Good agreement of both solutions is observed in relation to the level of dynamic strength and in the range of material softening.

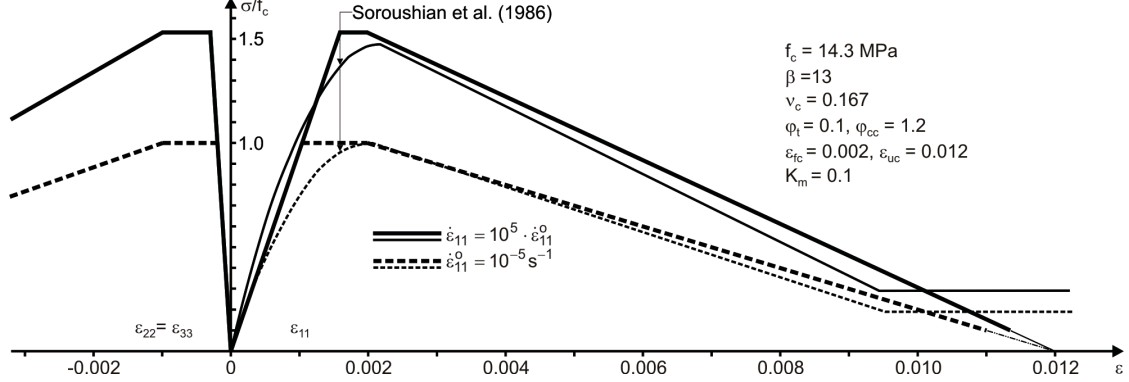

**Figure 16.** Comparison of the proposed model with theoretical model by Soroushian at al. [15].

## 4. Discussion

The presented model exhibits the following most important characteristic features.

1. Determination of the dynamic strength of concrete at any time-varying dynamic deformation process, using the universal integral criterion. This is—in the authors' opinion—the main advantage of this model.
2. Description of the spatial stress state, by means of a limit function, steered by the one parameter describing the homothetic expansion in the process of reaching the dynamic strength or shrinkage in the material softening process.
3. Combination of two qualitatively different stages of the deformation process: (1) process of reaching the dynamic strength of concrete accumulated exclusively in the elastic range, and (2) quasi-static incremental elastic–plastic process at the level of determined dynamic strength.
4. Estimation with the surplus of the elastic deformation range.
5. Omission of the influence of strain rate on the increase of dynamic deformation modulus, which partially reduces the effect of the stiffness from increasing of material described in the previous point.

6. Consideration of the degradation effect of the deformation modulus in the range of material softening.

7. Description of cracking or crushing states of concrete as the stress-less states reached in the material softening processes at tension or compression.

The comparative analysis shows good compatibility of the model with the experimental results and other models known from the literature, both in terms of dynamic strength description and description of behavior in the range of material softening.

The features of the presented model indicate that it can be used especially in the dynamic analysis of reinforced concrete structures.

The model has possibilities for further potential modifications both in the range of selection of material constants in the dynamic strength criterion based on the new results of dynamic experimental tests as well as in the range of determination of limit strains of material softening, therein for the high-strength concretes.

The work presents the monotonic behavior of concrete, essentially for uniaxial compression processes. The current direction of the authors' researches is to demonstrate the full possibilities of the model, illustrating the alternating processes of spatial, two- and three-dimensional stress states type of compression–tension–compression, showing the effects of cracks opening and reclosing.

## 5. Conclusions

A model of the dynamic behavior of concrete was presented in the paper. The proposed model describes a four-phase approximation of the nonlinear behavior of concrete.

The assumed range of physical non-linearity of the constitutive equation for the concrete traces the effects of reaching the dynamic strength, perfectly plastic flow, material softening and the cracking or crushing in the regions of the critical stressing/straining of the structural material. The linear elastic phase terminates when the yield surface of concrete is reached. This surface is non-classically described by the dynamic strength coefficient calculated on the base of the integral strength criterion for concrete. Perfectly plastic and material softening phases are determined by the global limit strains values. The changes of the yield surface in the stress space are controlled by the evolution parameter. The cracking and crushing mechanisms are interpreted as the processes, which result from the assumed material softening rule.

The proposed model may have a potential application in modeling the dynamic behavior of both concrete (with a lower level of accuracy) and reinforced concrete elements and structures (with a high degree of accuracy). Modeling of static concrete behavior and is also possible assuming the omission of determination of dynamic strength of concrete.

Model parameters determining its effectiveness are given in the paper. Namely, they are as follows: static strengths of concrete for uniaxial compression, uniaxial tension, and biaxial symmetrical compression, initial modulus of elasticity, material constant in degradation parameter of modulus of elasticity, transverse strains (Poisson's) coefficient, material constants in dynamic strength criterion, limit strains for the perfectly plastic flow phase and for the material softening phase, minimum value of the evolution parameter determining the end of the material softening phase, material constant defining the plastic potential function.

The most relevant limitation of the model is the assumption of linear idealizations for the elastic range of deformation until the instant of achieving the dynamic strength as well as for the perfect plasticity and the material softening deformation phases. The most significant advantage of the model is the ability to determine the dynamic strength of concrete for any variable in time deformation process as well as the ability to describe the cracking and crushing states of concrete.

**Author Contributions:** Conceptualization, A.S. (Adam Stolarski); methodology, A.S. (Adam Stolarski); software, A.S. (Adam Stolarski) and W.C.; validation, A.S. (Adam Stolarski) and W.C.; formal analysis, A.S. (Adam Stolarski), W.C. and A.S. (Anna Szcześniak); investigation, A.S. (Adam Stolarski), W.C. and A.S. (Anna Szcześniak); resources, A.S. (Adam Stolarski) and A.S. (Anna Szcześniak); data curation, A.S. (Adam Stolarski) and W.C.; writing—original draft preparation, A.S. (Adam Stolarski); writing—review and editing, A.S. (Adam Stolarski), W.C. and A.S. (Anna Szcześniak); visualization, A.S. (Adam Stolarski) and W.C.; supervision, A.S. (Adam Stolarski); project administration, A.S. (Adam Stolarski) and W.C.

**Funding:** This research received no external funding.

**Acknowledgments:** The research was supported by the Statutory Research Grant no 934 in the Faculty of Civil Engineering and Geodesy, Military University of Technology, Warsaw, Poland.

**Conflicts of Interest:** The authors declare no conflicts of interest.

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
