# Peer review of "Non-Classical Model of Dynamic Behavior of Concrete"

_applsci, doi:10.3390/app9132590_

Round 1
Reviewer 1 Report
The reviewer appreciates for the author's efforts for fulfilling the research work on the presented paper. The presented paper was aimed to develop the three-dimensional elasto-plastic consitutive model for dynamic respponse of concrete, and presents the reserach works from theory to practical applcations to concrete dynamic response under various loadng paths. The contens and developments contained in the presente paper are valuable in investigating dynamic response of concrete and worthy of publishing in the Journal as a research paper. However, even if the paper is well written and organized, some improvements are needed in some aspects of expressions and concepts:
1 "Introduction" illustrates many research papers related to the presented research work that are mostly adopted for the theoritical developments and varifications of the proposed model. However, the most of literature reviews are just illustrations of the research works performed before the presented work. More comments and explanations are needed to interrelate the works of the references with the presented work. Also "Introduction" needs to include the necessity, purpose, approch methos within the presented work.
2. In line 191, please correct the miswriting "on the of static..."
3. In line 207, the expression "Determination the dynamic stength coefficient..." is not smooth, Maybe "Determination of the dynamic..."
4. In line 211, the expression "further extension this asumption onto any sress state..." is not smooth, Similaly to 3.
5. In line 228, the expression " strain rates decomposition..." is not smooth, Maybe "strain rate decemposition..."
6. The paper adopts non-associated flow rule to describe the dilatant effect of pressure sensitve concrete material under confining pressure by introducing the parameter β into equation (1). In Eq. (16), please add more explanations for the role of β in changing the direction of plastic flow vector especially under confining pressure.
7. In Figure 2, the vertical axis has no name.
8. Sectin 2.8 introduces degradation of elastic modulus under low-cycle experiment by Sinha et al. Does the model include the degradation of elastic modulus written in (26) like damage-plasticity concrete model? If so, the paper needs to explain the derivation process soving multi energy dissipative mechnisms with more than two state variables. Please state it in the paper how the degradation of elastic modulus was incororated with elasto-plasticity.
9. Section 2.9 is an elementary procedure for plastic-flow based elastoplastic formulation. This part maybe shortened by removing well-known equations of (3), (33), (34), (35), (36), .. depending on the author's view points.
10. In line 351, 360, no Figure 5(c), 5(e), 5(f)
11. Figure 4 are not stated in the paper, probably Figure 5 of 10 to Figure 4
12. In line 357 to 358, the expression "its capability to repeated commencement of the compression..." is not smooth, maybe "to repeat commencement of..."
13. In line 376 and 390, Section 3.1 are doubled.
14. The developed model is full three-dimensional elasto-plastic model that features the hydrostatic as well as deviatoric responses by adoting the Willam-Warnke C-1 contnuity parabolic function of deviatoric length. However, practical applications are made in uniaxial experiement and old version of experimenental resulus of Kupfer et al. on two-dimansional plane stress. The reviewer would like to recommend the authors to make applications to three-dimensional experimental results such as (Smith et al. ACI Materials. J, V.86. 1989)
Author Response
Professor Adam Stolarski, Ph. D, D. Sc.
Military University of Technology
Faculty of Civil Engineering and Geodesy
2 gen. Sylwestra Kaliskiego Street
00-908 Warsaw-49, Poland
16 June, 2019
MDPI AG
Applied Sciences
Publication Title: Non-classical Model of Dynamic Behavior of Concrete
Journal Title: Applied Sciences
Manuscript Authors: Adam Stolarski, Waldemar Cichorski, Anna Szcześniak
To Reviewer 1:
I would like to thank you very much for insightful review of our manuscript. I am sure that it influents positively for final shape of our manuscript. We have considered and have taken into account all of your remarks during correction of the manuscript as follow. Moreover, all changes have been marked in the manuscript as comments.
Remark 1
"Introduction" illustrates many research papers related to the presented research work that are mostly adopted for the theoretical developments and verifications of the proposed model. However, the most of literature reviews are just illustrations of the research works performed before the presented work. More comments and explanations are needed to interrelate the works of the references with the presented work. Also "Introduction" needs to include the necessity, purpose, approach methods within the presented work.
· Taken into account as described in Comments [AS2] and [AS3] in the text of the work.
Remark 2
In line 191, please correct the miswriting "on the of static..."
· Taken into account as described in Comment [AS4] in the text of the work.
Remark 3
In line 207, the expression "Determination the dynamic strength coefficient..." is not smooth, Maybe "Determination of the dynamic..."
· Taken into account as described in Comment [AS5] in the text of the work.
Remark 4
In line 211, the expression "further extension this assumption onto any stress state..." is not smooth, Similarly to 3.
· Taken into account as described in Comment [AS6] in the text of the work.
Remark 5
In line 228, the expression " strain rates decomposition..." is not smooth, Maybe "strain rate decomposition..."
· Taken into account as described in Comment [AS8] in the text of the work.
· Title of section 2.5 have been also corrected.
Remark 6
The paper adopts non-associated flow rule to describe the dilatant effect of pressure sensitive concrete material under confining pressure by introducing the parameter β into equation (1). In Eq. (16), please add more explanations for the role of β in changing the direction of plastic flow vector especially under confining pressure.
· Taken into account as described in Comment [AS7] in the text of the work.
Remark 7
In Figure 2, the vertical axis has no name.
· Taken into account as described in Comment [AS9] in the text of the work.
Remark 8
Section 2.8 introduces degradation of elastic modulus under low-cycle experiment by Sinha et al. Does the model include the degradation of elastic modulus written in (26) like damage-plasticity concrete model? If so, the paper needs to explain the derivation process solving multi energy dissipative mechanisms with more than two state variables. Please state it in the paper how the degradation of elastic modulus was incorporated with elastic-plasticity.
· Taken into account as described in Comment [AS10] in the text of the work.
Remark 9
Section 2.9 is an elementary procedure for plastic-flow based elastic-plastic formulation. This part maybe shortened by removing well-known equations of (3), (33), (34), (35), (36), .. depending on the author's view point.
· Considered as described in Comment [AS11] in the text of the work.
Remark 10
In line 351, 360, no Figure 5(c), 5(e), 5(f)
and
Remark 11
Figure 4 are not stated in the paper, probably Figure 5 of 10 to Figure 4
· Taken into account as described in Comments [AS12] – [AS14], [AS16] and [AS17] in the text of the work.
Remark 12
In line 357 to 358, the expression "its capability to repeated commencement of the compression..." is not smooth, maybe "to repeat commencement of..."
· Taken into account as described in Comment [AS15] in the text of the work.
Remark 13
In line 376 and 390, Section 3.1 are doubled.
· Taken into account as described in Comment [AS19] in the text of the work.
· Number of the next section have been also changed - Comment [AS23].
Remark 14
The developed model is full three-dimensional elastic-plastic model that features the hydrostatic as well as deviatoric responses by adopting the Willam-Warnke C-1 continuity parabolic function of deviatoric length. However, practical applications are made in uniaxial experiment and old version of experimental results of Kupfer et al. on two-dimensional plane stress. The reviewer would like to recommend the authors to make applications to three-dimensional experimental results such as (Smith et al. ACI Materials. J, V.86. 1989)
· Considered as described in Comment [AS25] in the text of the work.
Once again we would like to thank you very much for all remarks that gave us opportunity to improve our manuscript.
Sincerely,
Adam Stolarski

Reviewer 2 Report
General comment
The character of the paper is theoretical since Authors did not provide any results from practical applications of the delivered model. The scientific aspects of the research are clearly underlined. The quality of the manuscript is rather good, and the presentation clear. The research methodology is correct. However, some modification of the manuscript could increase its quality.
Specific comments:
[1] Line 20: Please change the ‘very effective’ for more accurate expression.
[2] It seems to be good to modify the Introduction section. There are a lot of literature references that are not explained enough. Is any similarity in the delivered model of concrete to the concrete damage plasticity model (CDP)? If yes, please clarify this issue. Please also delivers information about the novelty of the paper and its potential application.
[3] The Discussion section (in this form) looks like a list of short remarks with no explanation. Please consider the more readable form of the Discussion section.
[4] In the Conclusion section, could you consider adding any information about the potential application of the presented the non-classical model of concrete? There are any limitations? If it is possible, it will be very helpful to provide information about parameters for the model (for calculations).
Author Response
Professor Adam Stolarski, Ph. D, D. Sc.
Military University of Technology
Faculty of Civil Engineering and Geodesy
2 gen. Sylwestra Kaliskiego Street
00-908 Warsaw-49, Poland
16 June, 2019
MDPI AG
Applied Sciences
Publication Title: Non-classical Model of Dynamic Behavior of Concrete
Journal Title: Applied Sciences
Manuscript Authors: Adam Stolarski, Waldemar Cichorski, Anna Szcześniak
To Reviewer 2:
I would like to thank you very much for insightful review of our manuscript. I am sure that it influents positively for final shape of our manuscript. We have considered and have taken into account all of your remarks during correction of the manuscript as follow. Moreover, all changes have been marked in the manuscript as comments.
Remark 1
Line 20: Please change the ‘very effective’ for more accurate expression.
· Taken into account as described in Comment [AS1] in the text of the work
Remark 2
It seems to be good to modify the Introduction section. There are a lot of literature references that are not explained enough. Is any similarity in the delivered model of concrete to the concrete damage plasticity model (CDP)? If yes, please clarify this issue. Please also delivers information about the novelty of the paper and its potential application.
· Taken into account as described in Comments [AS2] and [AS3] in the text of the work.
Remark 3
The Discussion section (in this form) looks like a list of short remarks with no explanation. Please consider the more readable form of the Discussion section.
· Considered as described in Comment [AS24] in the text of the work.
Remark 4
In the Conclusion section, could you consider adding any information about the potential application of the presented the non-classical model of concrete? There are any limitations? If it is possible, it will be very helpful to provide information about parameters for the model (for calculations).
· Taken into account as described in Comment [AS26] in the text of the work.
Once again we would like to thank you very much for all remarks that gave us opportunity to improve our manuscript.
Sincerely,
Adam Stolarski
